# Unraveling the Anticancer Potential of Statins: Mechanisms and Clinical Significance

**DOI:** 10.3390/cancers15194787

**Published:** 2023-09-29

**Authors:** Mohamed Y. Zaky, Chuanwen Fan, Huan Zhang, Xiao-Feng Sun

**Affiliations:** 1Department of Oncology, Linköping University, 581 83 Linköping, Sweden; 2Department of Biomedical and Clinical Sciences, Linköping University, 581 83 Linköping, Sweden; 3Molecular Physiology Division, Zoology Department, Faculty of Science, Beni-Suef University, Beni-Suef 62521, Egypt

**Keywords:** statins, anticancer activities, signaling pathways, cancer therapy, clinical application

## Abstract

**Simple Summary:**

Statins are crucial for managing lipid disorders by inhibiting HMG-CoA reductase, reducing cholesterol levels, and lowering cardiovascular disease risk. Beyond cholesterol control, they exhibit pleiotropic effects, particularly in cancer. Statins influence key cancer pathways, inhibiting proliferation, angiogenesis, metastasis, and cancer stemness, while inducing oxidative stress, cell cycle arrest, autophagy, and apoptosis. Clinical studies suggest statin use associates with reduced cancer risk, lower-grade tumors at diagnosis, decreased local recurrence, and improved survival. This review aims to summarize the mechanisms underpinning statins’ anticancer properties and their clinical implications, highlighting their potential as a valuable tool in cancer prevention and treatment.

**Abstract:**

Statins are an essential medication class in the treatment of lipid diseases because they inhibit 3-hydroxy-3-methylglutaryl-coenzyme A (HMG-CoA) reductase. They reduce cholesterol levels and reduce the risk of cardiovascular disease in both primary and secondary prevention. In addition to their powerful pharmacologic suppression of cholesterol production, statins appear to have pleitropic effects in a wide variety of other diseases by modulating signaling pathways. In recent years, statins have seen a large increase in interest due to their putative anticancer effects. Statins appear to cause upregulation or inhibition in key pathways involved in cancer such as inhibition of proliferation, angiogenesis, and metastasis as well as reducing cancer stemness. Further, statins have been found to induce oxidative stress, cell cycle arrest, autophagy, and apoptosis of cancer cells. Interestingly, clinical studies have shown that statin use is associated with a decreased risk of cancer formation, lower cancer grade at diagnosis, reduction in the risk of local reoccurrence, and increasing survival in patients. Therefore, our objective in the present review is to summarize the findings of the publications on the underlying mechanisms of statins’ anticancer effects and their clinical implications.

## 1. Introduction

Statins, or HMG-CoA reductase inhibitors, were discovered in the 1970s by the Japanese researcher Akira Endo [1]. Statins have significantly improved the therapeutic management of excessive cholesterol and ischemic heart disease since their clinical introduction in the late 1980s, and their usage has grown ubiquitously worldwide. Statins are specific inhibitors of the mevalonate (MVA) pathway that is involved in the de novo synthesis of cholesterol and other nonsterol isoprenoids. HMG-CoA reductase (HMGCR) is the rate-limiting enzyme in MVA synthesis [2]. Statins are effective in the treatment of hypercholesterolemia because they inhibit HMGCR [3]. There is a great deal of interest in repurposing statins as anticancer medications for both cancer prevention and treatment. As a result, the anticancer mechanisms of statins have received much interest. The potential benefit of statins against cancer has grown in importance as they are significantly reducing the risk of developing different types of cancer. Many studies using cancer cell lines from various epithelial origins, including skin, breast, prostate, ovarian, liver, colorectal, and lung cancer, have indicated that prolonged statin exposure effectively inhibits the growth, proliferation, and induced apoptosis of these cancer cells [3]. Further, statins have been utilized in combination with other chemotherapeutic medications to impede cell proliferation and trigger apoptosis. Moreover, many in vivo studies have demonstrated the anticancer therapeutic impact of statins by suppressing the tumor size and inhibiting migration and proliferation. Advanced clinical trials are currently being conducted with numerous encouraging in vitro and in vivo results on the effectiveness of statin therapy in several cancer types. These studies have found that statin use is associated with a decreased risk of cancer formation, a lower cancer grade at diagnosis, and a lower risk of cancer recurrence and death [4]. The discovery of novel anticancer drugs is progressing, but rising medical costs associated with drug research have evolved into a significant obstacle. Statins, which are widely available, affordable, and well tolerated, have yet to be successfully repurposed for cancer treatment [5]. In this article, we reviewed the potential anticancer mechanisms of statins and their clinical implications in cancer therapy.

## 2. Anticancer Mechanisms of Statins

### 2.1. The Mevalonate Pathway

Cancer cells exhibit characteristic metabolic alterations. However, finding medications that target various tumor types and cancer-specific metabolic dependency has always been difficult. The MVA pathway is a crucial and intricate signaling mechanism that controls the synthesis of several isoprenoids, including polyol, ubiquinone, vitamin D, cholesterol, and lipoproteins [6]. While the MVA pathway which is a fundamental requirement for all cells, it has also been found to be elevated in all carcinogenic lesions, according to mounting evidence [7]. An increase in MVA demand is a marker of carcinogenesis, and because there are more MVA pathway intermediates available, tumor cells are more resistant [8]. The existence of selective and well-tolerated inhibitors makes the MVA pathway a prime target for cancer treatment. Millions of people receive statin prescriptions for the treatment of hypercholesterolemia because they prevent HMGCR from producing MVA. There are two types of impacts that can be distinguished: those caused by cholesterol-mediated pathways and those caused by non-cholesterol-mediated pathways.

#### 2.1.1. Non-Cholesterol-Mediated Pathways

Acetyl-CoA, the precursor to MVA, is produced by decarboxylating pyruvate, the end result of glycolysis. The rate-limiting step in the MVA pathway is the process where 3-hydroxyglutaryl-CoA is transformed into MVA by HMGCR from tree molecules of acetyl CoA [9]. MVA is subsequently phosphorylated by MVA kinase and converted to isopentenyl pyrophosphate (IPP), which outlines the fundamental role played by the MVA pathway in the processes that farnesyl diphosphate synthase (FDPS) catalyzes to produce farnesyl pyrophosphate (FPP) and geranylgeranyl pyrophosphate (GGPP) [10]. Squalene, a precursor to cholesterol and a crucial byproduct of the MVA pathway, is produced using FPP as its basic building block. Squalene epoxidase and squalene synthase are two more enzymes that modulate cholesterol [11]. Cholesterol plays a crucial role in the development and maintenance of cellular membrane structure and function. Additionally, it serves as a precursor for bile acids, vitamin D, and steroid hormones [12]. The synthesis of FPP and GGPP is necessary for the prenylation of many proteins as well as a post-translational modification. The enzymes farnesyltransferase (FTase) and geranyltransferases (GGTases) I and II mediate this process. Prenylation that occurs post-translationally is necessary for the membrane anchoring, localization, and activity of several signaling proteins [9]. These proteins, which participate in a number of essential intracellular signaling pathways, include the subunit of G-protein-coupled receptors (GPCRs) and tiny monomeric GTPases (guanosine-triphosphate hydrolase). These small monomeric GTPases regulate a number of cellular processes, including endocytosis/exocytosis, differentiation, migration, apoptosis, and proliferation through isoprenylation and the association of these signaling molecules with effectors [9]. Cellular MVA, IPP, FPP, and GGPP pools become reduced as a result of statins’ suppression of HMGCR. RAS and Rho family GTPases are affected by this process [7]. The main functional effects of statins in the cancer treatment process are decreased GGPP and FPP concentrations, which inhibit RAS and Rho isoprenylation, signal transduction, and DNA synthesis [13]. These proteins’ activities are constrained, and their farnesylation may prevent the proliferation of cancer cells [14].

#### 2.1.2. Cholesterol-Mediated Pathways

A century ago, the initial indication of a connection between cancer and cellular cholesterol levels was documented in a publication [15]. From that time onward, a number of investigations have revealed that cholesterol levels in tumors are higher than in normal tissues [16,17]. It has been noted that tumor cells utilize the inadequacy of effective feedback regulation concerning LDL to increase intracellular cholesterol [18]. Additionally, cholesterol plays a crucial role in lipid rafts and is crucial for cell growth and the cell cycle, particularly for the transition from G1 phase to S phase [19]. To address the needs of tumor-promoting cell signaling proteins, cancer cells have a greater need for cholesterol and contain more lipid rafts than normal cells [20]. Through the MVA pathway, cells can de novo synthesize cholesterol or increase cholesterol uptake from the plasma through LDLR-mediated endocytosis. The identification of sterol regulatory element-binding proteins (SREBPs) represents a major advance in our comprehension of the control of MVA pathway genes. These transcription factors are created on the endoplasmic reticulum (ER) membrane as inactive precursors, where they attach to sterol sensors called SREBP cleavage-activating proteins (SCAPs). When intramembranous cholesterol levels drop, SCAPs relocate SREBP-2 to the Golgi apparatus. High intracellular cholesterol levels cause SCAPs to block SREBP translocation and activation, which inactivates HMGCR and LDLR transcription [21]. The intricate control over these regulatory mechanisms makes sure that cells’ cholesterol homeostasis is maintained [6]. Dysregulation of cholesterol homeostasis or important components in cholesterol is associated with the emergence of cancer via inflammasome- and micRNA-mediated pathways in addition to numerous well-known carcinogenic pathways. SREBP2 also controls the transcription of genes involved in NOD-like receptor protein 3 (NLRP3)-related inflammation. It is well documented that persistent inflammation and cancer are causally related. The proinflammatory cytokines interleukin (IL)-1 and IL-18, as well as the inflammatory protease caspase-1, are all activated by inflammasomes, which are created in response to inflammation [22]. The growth and progression of malignancies are closely correlated with the regulation of NLRP3 [23,24,25]. To favorably control the expression of SREBP2, microRNA 33 (miRNA33) interacts with the SREBP2 gene. Uncontrolled cell division is brought on by hyperactivation of cholesterol biosynthesis [26]. Cholesteryl acyltransferase 1 (ACAT1) esterifies excess cholesterol to create cholesteryl esters (CEs), which are then stored in cells as lipid droplets (LDs), in order to prevent an excessive buildup of free cholesterol [18].

Cholesterol transfer protein carries freshly created free cholesterol to subcellular membranes. Although CEs act as a reservoir for cholesterol, their buildup or overexpression by ACAT1 enhance carcinogenesis [6]. ACAT1 ablation has been found to inhibit tumor growth in a glioblastoma xenograft model [27]. Hepatocellular carcinoma (HCC) was also shown to have ACAT1 overexpression [28]. Additionally, it was observed that inhibiting the expression of ACAT1 reduced the growth and migration of breast and prostate cancers [29]. Overexpressing Proprotein-Convertase-Subtilisin-Kexin Type-9 (PCSK9) promotes the lysosomal degradation of LDLR [30], which causes hypercholesterolemia and eventually HCC [31]. Additionally, extra cholesterol creates oxysterols, which are organic ligands for liver X receptors (LXRs). The interaction between the co-activator protein and the receptor is improved when cholesterol is bound to LXRs. This also triggers the transcription of genes related to cholesterol efflux, such as ATP-binding cassette (ABC) subfamily A member 1 (ABCA1), ABC subfamily G member 1 (ABCG1), and ABCG5/8. High-density lipid proteins (HDLs) are produced when extra cholesterol is transferred to apolipoprotein A-I (ApoA-I), which is a lipid-poor protein and is transported back to the liver [32]. The LXR can increase ABCA1 transcription when intracellular cholesterol levels are high [32]. However, the ABCA1 expression is suppressed in cancer cells by the phosphatidylinositol-3-kinase (PI3K), protein kinase B (AKT), and mechanistic target of rapamycin complex 1 (mTORC1) pathway [26]. MiR-183 directly degrades ABCA1 mRNA to maintain high amounts of intracellular cholesterol, which helps colon cancer cells proliferate and have antiapoptotic effects [33]. Similarly, MiR-27a-3p also prevents cancer cell death by inhibiting cholesterol efflux or by targeting ABCA1 [34]. It has been demonstrated that LXR overexpression inhibits the proliferation of gastric cancer cells [35]. The smoothened receptor (SMO), a GPCR that binds cholesterol and its oxygenated derivatives with high affinity, activates the Sonic Hedgehog (SHH) pathway [36]. By increasing the activity of the glioma-associated oncogene homolog 1 (GLI1), which in turn activates the hedgehog target genes and promotes tumor growth, the SHH pathway is thought to work as an oncogenic signaling cascade that can speed up the cell cycle and stem cell proliferation [37]. Statins suppress tumor growth via lowering cholesterol production and suppressing SHH signaling in fibroblasts and medulloblastoma cells [38]. This is corroborated by the finding that statins prevent medulloblastoma growth without damaging bone by preventing HH signaling in tumor cells [39].

### 2.2. Statins Regulate Autophagy

The biological processes of autophagy and apoptosis are both crucial and play a role in a variety of biological functions, including cell growth and differentiation. The main difference between autophagy and apoptosis is their role for the cells. For instance, autophagy primarily targets specific harmful components in stressful situations to ensure cell survival [40]. Autophagic cell death, which is analogous to apoptosis, occurs when autophagy is insufficient to alleviate stress and make up for cellular damage (Figure 1) [41]. Many investigations demonstrated that autophagy contributes to the development of pathological diseases, particularly cancer [42]. It has been revealed that autophagy inhibition of cancer cells can prevent cancer or reduce the viability of malignant cells. Hence, it appears that autophagy plays a crucial role in cancer growth [43]. Autophagy plays a key role in the tumor-suppressive process during the early stages of cancer growth, acting as a crucial quality control system to protect genomic integrity by controlling the breakdown of harmed proteins or organelles (such as malfunctioning mitochondria) [43]. Additionally, autophagy promotes senescence brought on by oncogenes [44], and aids in the detection of cancer by the immune system [45]. Autophagy mostly has tumor-promoting effects in the later phases of tumor growth. The dietary needs for anabolic pathways are high because cancer cells multiply quickly. By recycling cellular substrates, autophagy aids cancer cell metabolism (Figure 1) [46]. Under stressful circumstances like tumor hypoxia and ER stress, autophagy also helps cancer cells growth [47].

Several signaling pathways are implicated in the regulation of autophagy (Figure 2). The ability to regulate autophagy has contributed to the discovery of many anticancer drugs. Simvastatin, a hydrophobic statin, was reported to enhance autophagy in Rhabdomyosarcoma A204 cells by depleting the GGPP pool, according to the first study on statin-induced autophagy [48]. Fuvastatin hinders bone metastases in lung cancer cells (SPC-A-1) by promoting autophagy after the deletion of autophagy-related genes (ATG5 or ATG7) [49]. By blocking GGPP production, atorvastatin induces autophagy in HEK293 human embryonic kidney cells via the mevalonate pathway (Figure 3) [50]. Simvastatin’s effect on autophagy in U251 cells was inhibited by the addition of mevalonate, suggesting that the mevalonate pathway is involved in statin-mediated autophagy [51]. Instead of the formation of FPP or GGPP, the stimulation of autophagy flux in human leukemia cells treated with simvastatin for 24–72 h appears to be influenced by the suppression of cholesterol formation [52]. Activating autophagy without using the MEV pathway was achieved by atorvastatin in the Huh7 and HCT116 gastrointestinal cancer cell lines [53]. Simvastatin activates the extracellular regulated protein kinases 1 and 2 (ERK1/2) and Akt pathways, which suppresses autophagy and promotes cell death in breast cancer cells (Figure 3) [54]. Lovastatin dramatically decreases the capacity of malignant pleural mesothelioma tumor cells to survive and migrate by promoting autophagy [55]. As a target of cancer treatment, activating autophagic cell death may be preferred over cell death through apoptosis since tumor cells are resistant to apoptosis-based therapy and cancer cell apoptosis stimulates the survival and proliferation of nearby cells. The combination of lovastatin and farnesyltransferase inhibitors has been shown to increase nonapoptotic cell death and inhibit autophagy flux [56]. Apoptosis is induced and angiogenesis is inhibited by large dosages of atorvastatin. In order to increase autophagy, atorvastatin increases the expression of the light chain 3-phosphatidylethanolamine conjugate (LC3II), which decreases cancer cell survival and proliferation [57]. It has been demonstrated that the combination of lovastatin and cisplatin elevates LC3B-II expression and reduces cancer cell viability by causing autophagic cell death [58]. The emergence of chemotherapeutic resistance has been linked to autophagy [59]. Simvastatin impedes the formation of autophagolysosomes, thereby reducing the autophagic flux triggered by Temozolomide (TMZ) and heightening the vulnerability of glioblastoma cells to TMZ-induced cell death [60].

### 2.3. Statins Induce Ferroptosis

In contrast to apoptosis, necrosis, pyroptosis, and autophagy, ferroptosis is a type of programmed cell death (PCD) that was first described in 2012 [61]. Ferroptosis has been associated with many lethal diseases [62] and is gaining attention due to its possible application in the treatment of cancer [63]. Ferroptosis can be inhibited through strategies such as sequestering free iron, reducing the synthesis of polyunsaturated fatty acids (PUFAs), or scavenging reactive oxygen species (ROS) [64]. Statins cause ferroptosis through the MVA pathway, which is associated with the regulation of the GSH/GPX4 and FSP1/CoQ10/NAD (P) H axis. The MVA pathway is essential for GPX4 synthesis and the formation of the CoQ10 backbone. IPP, which is formed through the MVA pathway, is a precursor of CoQ10. IPP positively regulates Sec-tRNA, which plays an important regulatory role in the development of GPX4 [65]. Statins impair GPX4 translation efficiency and thereby predispose cells to ferroptosis by blocking the rate-limiting enzyme in the MVA pathway (Figure 4) [66]. It was recently discovered that the mechanism underlying CoQ10’s protective action is based on FSP1’s ability to employ CoQ10 as a substrate to inhibit lipid autoxidation [67].

### 2.4. Statins Induce Pyroptosis

Pyroptosis is a form of inflammatory PCD that differs from both ferroptosis and autophagy, with autophagy being distinct from pyroptosis. It was found in myeloid cells that had bacterial or disease infections in 1992 [68]. Pyroptosis is characterized by the release of proinflammatory molecules such as IL-1 and IL-18, cell swelling, rupture, and lysis [69] and brought on by members of the gasdermin superfamily such as GSDMA, GSDMB, GSDMC, GSDMD, and GSDME [70]. The inflammasome stimulates proteins from the caspase family to cleave gasdermin. The protein’s active form transfers to the cell membrane and creates holes that result in pyroptosis [71]. Inflammatory caspases (caspase 1/4/5/11), which are triggered by inflammasomes, cleave GSDMD to generate pyroptosis, whereas apoptotic caspases (caspase 3) cut GSDME to cause pyroptosis [72]. Pyroptosis primarily involves two main mechanisms: the caspase-1-mediated canonical pathway and the caspase-4/5/11-mediated noncanonical pathway. Inflammasomes, which are recruited and activated by caspase-1, which in turn promotes inflammatory molecules like IL-18 and IL-1 and cleaves the N-terminal portion of GSDMD, are used in the traditional pathway to identify damage. The cell membrane is then attached to by the GSDMD’s active form, which results in pyroptosis and the development of holes [73]. In the noncanonical pathway, human homologs caspase-4 and caspase-5, as well as murine caspase-11, recognize and bind to bacterial lipopolysaccharide (LPS). Subsequently, they cleave GSDMD, leading to the activation of caspase-1 and ultimately triggering pyroptosis [74].

Several studies have discovered a direct connection between pyroptosis and the onset and development of various diseases, including cancer. Zhou et al. [75], for example, revealed that to maximize the antitumor effects of therapeutic ROS-inducing drugs and to decrease melanoma cell growth and spread via GSDME-dependent pyroptosis, iron supplementation at the recommended levels in iron-deficient patients is sufficient. Atorvastatin inhibits pyroptosis in human vascular endothelial cells by inhibiting the long noncoding RNA (lncRNA) NEXNAS1/NEXN pathway [76]. Rosuvastatin was observed to decrease the levels of NLRP3, caspase-1, interleukin-1, and GSDMD N-terminal domains expression. This finding suggests that the medication has the potential to mitigate cardiac injury by reducing the occurrence of pyroptosis [77].

### 2.5. Statins Target the Tumor Microenvironment

The noncancerous cells, materials, and activities found in and around tumors are referred to as the “tumor microenvironment” (TME). As of now, it is understood that cancer is an ecological and evolutionary process that involves dynamic, continuing interactions between cancer cells and TME [78]. The ongoing interaction between tumor cells and the TME plays a crucial role in the development, metastasis, and response to treatment of tumors. TME, to variable degrees, aids in the development and maintenance of cancer hallmarks like immune evasion, sustained proliferative signals, sustained proliferative signals, and others [6]. The emphasis of cancer therapy has changed from a cancer-centric paradigm to a TME-centric one as a result of improved understanding of the TME’s crucial roles in tumor growth and therapeutic resistance. Targeting TME components has been attempted to treat cancer patients therapeutically [6]. It seems that a focused microenvironment is a more effective cancer treatment target than the TME as a whole. It has been shown that some effects of traditional antitumor medications are mediated through TME targeting. Statins have been demonstrated to particularly target these specialized microenvironments, resulting in antitumor actions [6]. In a mouse model of head and neck squamous cell carcinoma (HNSCC), it has been shown that simvastatin induces metabolic reprogramming within the metabolic microenvironment. This reprogramming is characterized by a reduction in lactic acid production and increased resistance of cancer cells to monocarboxylate transporter 1 (MCT1) inhibitors [79]. Simvastatin regulates cholesterol-associated LXR/ABCA1 to repolarize tumor-associated macrophages (TAMs) and promote M2-to-M1 phenotypic switching of macrophages in the mechanical microenvironment, ultimately changing the TME and reducing epithelial–mesenchymal transition (EMT) [80]. Statins have also been demonstrated to influence immune checkpoints, cytokines, and chemokines in the immunological microenvironment. Statins decrease the release of CCL3, as well as IL-6 and CCL2 from mesenchymal stromal cells, which limits the ability of lung cancer cells to survive. This suggests that statins have the potential to be repurposed as medications that target the immunological TME [81]. Simvastatin suppresses the activation of mesenchymal stem cells in inflammatory breast cancer (IBC) through inhibiting IL-6 production [82]. Simvastatin can enhance CD8+ T cells’ ability to fight tumors by reducing cholesterol in the TME [83]. Simvastatin was also observed to greatly boost pancreatic tumor apoptosis in mice when combined with MEK inhibitors [84]. By investigating the Mutations and Drugs Portal (MDP), Taccioli et al. [85] discovered that statins and dasatinib work well together to inhibit YAP/TAZ in cancer cells. Iannelli employed both in vivo and in vitro models to demonstrate that through the regulation of the MVA pathway and the YAP axis, the combination of valproic acid and simvastatin has the capability to enhance the sensitivity of metastatic castration-resistant prostate cancer cells to docetaxel, potentially surmounting resistance to this drug [86].

## 3. Statin Anticancer In Vitro Studies

Numerous in vitro investigations have shown that simvastatin has antiproliferative effects on many cancer cell lines. According to these in vitro investigations, simvastatin suppresses cancer cell growth by causing apoptosis and slowing cell cycle progression via multiple cell signaling pathways (Figure 5) [87]. According to current research, the effects of statins are dependent on cell line, concentration, and time of exposure to the drug [87]. In addition, some studies have investigated the use of simvastatin in combination therapy [87,88].

According to a study conducted using cancer cells derived from various epithelial sources, including, prostate (LNCaP and PC-3), colon (Caco-2 and HCT-116), skin (SCC-M7 and SCC-P9), lung cancer cell lines (Calu-3 and Calu6) as well as breast (MCF7 and SKBr-3), simvastatin exhibited a more potent inhibition of cell growth in cells that were poorly differentiated [89]. Simvastatin’s effects are more pronounced in highly metastatic malignant tumor cells than in benign tumors with the same origin, according to other investigations. Perhaps this is because malignant tumors spread faster and need more isoprenoids from MVA to improve cell survival signaling [88]. MDA-MB-231 cells were treated with simvastatin at different concentrations (1–5 µM) for 48 h, and simvastatin significantly caused cell nuclei fragmentation and cell death. Simvastatin dramatically enhanced the amounts of ROS in these cells in a dose-dependent way, causing oxidative stress and DNA damage [90]. Additionally, low dosages of atorvastatin and cerivastatin (0.005–0.01 µM) enhance vascular endothelial cells to develop new vascular structures. On the other hand, higher doses of statins (0.05–1 µM) inhibit the growth of endothelial cells, decreasing the ability of factors to promote the formation of new vascular networks, and triggering cell death [91]. However, it has been found that statins had no impact on histone deacetylase (HDAC) activity in the following cells: HepG2, MDA-MB-231, BRIN-BD11, and THP-1 [92]. The utilization of simvastatin in conjunction with an HDAC class II inhibitor (MC1568) leads to a synergistic antiproliferative effect in colorectal cancer [93]. Additionally, inhibiting HDAC1 in cancer cell lines (CAL-27 and SACC83) may improve statins’ anticancer effect [94]. Simvastatin and valproic acid’s anticancer effects were examined by Iannelli et al. [86] in prostate cancer cell lines. They discovered that this combination made castration-resistant metastatic prostate cancer cells more susceptible to docetaxel. Mevastatin, in combination with the HDAC inhibitor LBH589, has demonstrated the ability to inhibit the growth of triple-negative breast cancer [95]. These results provide a promising direction for ongoing studies aiming to discover new cancer treatments. As a result, it is possible that combining statins with HDAC inhibitors would be far more effective than either treatment alone [96].

In vitro studies conducted on medulloblastoma cell lines indicate that simvastatin inhibits HMG-CoA reductase, an enzyme involved in the regulation of metalloproteinase synthesis through its influence on the production of GGPP [97]. It is important to understand that HMG-CoA reductase is essential for preserving cell viability, that its inhibition results in apoptosis, and that this has a negative impact on cell proliferation [98]. In vitro-cultivated human breast cancer cells displayed significant alterations in the matrix metalloproteinase-9 (MMP-9) levels [99]. Fluvastatin and simvastatin inhibited MMP-9 release in murine and human macrophages [100]. In research involving human microvascular endothelial cells (HMEC-1), cerivastatin was observed to dose-dependently decrease or entirely inhibit the production or synthesis of MMP-2 (Figure 5) [100]. Simvastatin effectively reduced cell viability by impairing GCTB stromal cell growth and causing apoptosis [101]. Additionally, simvastatin had the strongest effects on the reduction in AtT20 cell proliferation among all statins [102]. Furthermore, in nasopharyngeal carcinoma (NPC), simvastatin significantly reduced cell viability in C666-1 cells and enhanced apoptosis. By controlling the phosphorylation of the transcriptional factor c-Jun, simvastatin increased the expression of Bim. The activation of protein kinase B and extracellular signal-regulated kinase 1/2 was inhibited by simvastatin treatment (Figure 5) [83]. Also, Wang et al. [103] found that simvastatin caused pyroptosis in A549 and H1299 lung cancer cells, suggesting that simvastatin could be a suitable medication for lung cancer without harming healthy lung cells.

## 4. Statin Anticancer In Vivo Studies

Simvastatin’s anticancer efficacy has been demonstrated in published in vivo studies against a variety of tumor subtypes. Compared to dosages used in patients with hypercholesterolemia, supratherapeutic doses of simvastatin have been used in these investigations. Rodents need higher dosages of the medication to provide the same therapeutic effects because statins’ blood clearance and liver uptake are exaggerated [104]. Simvastatin inhibits pituitary neuroendocrine tumors from secreting hormones, which has antisecretory and antiproliferative actions [102]. Tumor cell growth was reportedly suppressed in a concentration-dependent way in studies using mice xenograft models for lung cancer and leiomyoma [104]. Simvastatin decreased tumor growth and stopped prostate cancer cells from proliferating, migrating, and invading, as demonstrated by Miyazawa et al. [105] in an examination of nude mice xenograft from PC-3 cells. Chen et al. [77] also demonstrated simvastatin’s anticancer effects on colorectal cancer, and the outcomes suggested that the medication may have anticancer properties. Li and Gan [94] showed that simvastatin pretreatment significantly delayed angiogenesis in the same tumor model and reduced microvessel density in vivo. Simvastatin, according to Kamel et al. [106] inhibited osteosarcoma growth in a way that was reliant on the MVP route suppression. Renno et al. [107] detected that simvastatin was found to have no effect on the stem cells in healthy, non-neoplastic breast tissues, but it significantly (by around 80%) decreased the formation of breast tumors at a dose of 40 mg/kg/day. Karimi et al. [108] also investigated the therapeutic effect of simvastatin on mice-induced breast cancer. They discovered that simvastatin increased breast carcinogenesis parameters including the typical tumor volume and the percentage of mortality in comparison to mice with untreated breast tumors. Furthermore, it appears superior in histological research when compared to mice given tamoxifen, a popularly recommended therapeutic treatment. In the ischemic limbs of normocholesterolemic rabbits, research has indicated that simvastatin can trigger the phosphorylation of endothelial nitric oxide synthase, prevent apoptosis, and support angiogenesis through an Akt-dependent mechanism [100]. Lipophilic statins may be more efficient at treating cancer than hydrophilic statins because they are more proapoptotic and have a higher potential for cytotoxicity [109].

Pitavastatin was administered intraperitoneally in mice during in vivo investigations, and this prevented the growth of subcutaneous glioma cells [100]. Simvastatin decreased bone metastasis and tumor growth in a mouse model of human lung cancer xenograft, and these effects were accompanied by a decrease in MAPK/ERK activity (Figure 3) [110]. Mice treated with a mixture of gemcitabine and fluvastatin also showed an inhibition and delaying of tumor growth in a pancreatic cancer xenograft [111]. With numerous encouraging in vitro and in vivo results on the effectiveness of statin therapy in numerous cancer models, advanced clinical trials are currently being conducted [112].

## 5. Clinical Application of Statins

### 5.1. Epidemiology

Numerous studies have been conducted on the subject of statins and cancer since the first reports of statins affecting cancer progression in the late 1990s, with varying degrees of success [110,111,112,113]. Statins have a well-established safety profile, making it more affordable to perform clinical trials on their usage in oncology than it is to develop novel medications [112]. Statins and CRC have been the subject of numerous epidemiological and clinical investigations. It has been reported that using statins for five or more years was associated with a 45% reduction in the risk of developing CRC (95%CI: 0.40–0.74) [114]. Another study found that taking statins reduced CRC risk by 35% (95% CI: 0.55–0.78) [115].

### 5.2. Meta-Analysis

An investigation into the utilization of statins in the field of oncology within the Danish population was included in one of the more extensive meta-analyses conducted recently [112]. Statin usage can lower cancer mortality by roughly 40%, according to meta-analysis [116]. In contrast, an alternative meta-analysis found that the simultaneous administration of statins alongside conventional anticancer drugs did not result in a notable enhancement of survival rates or progression-free survival among patients diagnosed with advanced cancer and a life expectancy of less than two years [117]. Researchers discovered that the usage of any type of statin was related to a decrease in the mortality of patients with pancreatic cancer in a retrospective cohort analysis (2006–2014) [112]. Findings by Ahmadi et al. [118] suggested that statin may increase the risk of developing specific cancers, such as melanoma and prostate cancer. The anticancer activity of statins is largely due to their ability to trigger apoptosis, which allows them to target cancer cells with excellent selectivity [119]. The MVA pathway appears to be key in the mechanisms behind their proapoptotic activity, which are linked to the inhibition of the cholesterol production pathway. Statins cause the MVA pathway’s intermediate metabolites to secrete less frequently, which contributes to the appearance of morphological changes that are indicative of programmed cell death [120]. Three different meta-analyses have looked into the relationship between statins and esophageal cancer. For example, Zhou et al. [121] reported that patients with esophageal cancer who took statins had a 26% better overall survival (95% confidence interval (CI): 0.75 to 0.94) and survival without disease (95% CI: 0.75–0.96). A total of 24,576 patients from five cohort studies were included in this meta-analysis. In another meta-analysis, in 1761 patients with pancreatic cancer, the 5-year overall survival rate was 16.6% for statin users and 8.9% for nonusers (*p* = 0.012). [122]. Among 226 patients who had their pancreatic cancer surgically removed, the active usage of moderate-to-high-dose simvastatin was connected to an enhanced OS and disease-free survival [122].

### 5.3. Observational Studies

One extensive observational study utilized data from the Danish Cancer Registry, making it one of the largest investigations into the role of statins in cancer prevention. Statins were discovered to significantly lower cancer incidence in a population of 300,000 patients (OR 0.86; 95% CI 0.78–0.95) [123]. According to a meta-analysis by Kuoppala et al. [124], which comprised 25 observational studies and 17 randomized trials, there was no discernible difference between the statin-taking group and the control group in terms of the risk of developing cancer (OR 0.96; 9% CI 0.72–1.20). It is interesting to note that a different meta-analysis based on 20 case–control studies showed statins to be effective in preventing hyperproliferative disorders [125]. These findings also examined the quality of the evidence used to support the results of 43 studies. Statins were found to have a positive effect in 10 out of the 18 tumors that were included, according to the standard method of significance assessment (*p* ≤ 0.05). Nevertheless, when the strength of the data was taken into consideration, statin use had no discernible impact on the chance of developing cancer. The advantages of statin medication should be assessed individually in each patient [88].

### 5.4. Randomized Controlled Trials (RCTs)

Long-term RCTs are crucial for research [126]. In one study, 20,000 cancer patients were randomly divided into two groups and given either a placebo or 40 mg of simvastatin for their cardiovascular effects [127]. An 11-year follow-up [128], that included 17,519 patients out of the 20,000 patients previously stated, did not show a relationship between statin use and the incidence of cancer or cardiovascular events; however, LDL concentration was observed to have decreased by 1 mmol/L over the observation period [129]. A meta-analysis of 26 randomized trials comprising 86,936 people, 6662 cancer diagnoses, and 2407 cancer-related deaths also found no evidence that statin use affected morbidity and survival, regardless of the types of cancers [130]. The results of the Cholesterol Treatment Trialists’ (CTT) study, which included an analysis of 27 randomized controlled trials (RCTs) involving a total of 174,149 individuals, reaffirmed that statin use did not show a significant impact on either cancer risk (odds ratio [OR] 1.00, 95% confidence interval [CI] 0.96–1.04) or survival (OR 0.99, 95% CI 0.93–1.06) [130]. However, the use of statins as adjuvant therapy for hormone-dependent breast cancer was shown to be effective in randomized studies, improving disease-free survival (HR 0.79; 95% CI 0.66–0.95) [130]. Regarding their role as adjuncts to traditional oncological therapies or in preventive medicine, it appears that well-known compounds like statins or metformin, which are well known for their low cost and their predictable adverse effects, may prove to be attractive. However, it takes carefully planned research to establish their efficacy [131]. Beta-ketoacyl reductase inhibitor (TVB-2640), another lipid-lowering drug, is undergoing clinical trials to determine its effectiveness and safety in treating human epidermal growth factor receptor 2 (HER2)-positive breast cancer. If yielding promising results, it is of interest for the study of other HER2-positive cancers, such as colon cancer and non-small-cell lung cancer. When coupled with bevacizumab, this drug increased life expectancy in glioblastoma patients relative to historical controls and decreased hepatic de novo lipogenesis in obese patients. De novo fatty acid production is also common in prostate cancer. However, TVB-2640 has not yet been investigated in this kind of cancer [132]. It can, in turn, suppress EMT in lung cancer patients in a way that is p53 mutation-dependent [133]. Studies show that statins inhibit class I and II HDACs and enhance histone H3 and H4 acetylation [134].

### 5.5. Pharmacoepidemiological Studies

The number of pharmacoepidemiological investigations has increased recently due to the easier availability of population data sets stored in repositories [135]. Through the simultaneous consideration and comparison of numerous factors, such analyses enable us to evaluate the true impact of medications, including statins, on the prevention and treatment of cancer [135]. An illustration of this approach is the AspECT study, which examined the efficacy of esomeprazole and aspirin in preventing the development of esophageal cancer in patients with Barrett’s esophagus at 84 facilities in the UK and one in Canada. The pharmacoeconomic evidence that was available supported the need for this investigation, and it was proven that the use of these two medications had a chemopreventive effect on esophageal cancer in patients with Barrett’s esophagus [136]. As drug effectiveness studies based on population registries may be hampered by flaws, such as the methodology, a lack of randomization, and bias [137], it would appear that this form of study should be conducted with caution. In a meta-analysis, Soni et al. [138] looked at the results of population studies and RCTs that were devoted to evaluating the efficacy of cancer treatments, but they were unable to find any agreement between them. According to Dickerman et al. [139], statin users have a considerably lower risk of developing cancer in several observational studies than in meta-analyses of randomized trials.

### 5.6. Mendelian Randomization

An important area of study is to examine the impact of genetic variations discovered in populations on the incidence of a certain effect by using Mendel’s randomization techniques. It is predicated on the widely accepted understanding that the polymorphism of the pertinent genes is connected to particular behaviors. The confounding variables and mistakes that come with conducting observational research are avoided in the study of genetic variants. The inclusion of randomization has ushered in a new era of genetic research. It can be used as a substitute for, or an addition to, RCTs to establish a causal link between exposure to a specific factor and the development of a disease. It is employed to discover relationships between gene families and the occurrence of disease. Natural randomization, according to Mendel’s second law, occurs because reproductive cells (sperm and ova) randomly disseminate gene variants (polymorphisms). On these occurrences, “Mendelian randomization” as a notion is built [133]. Research into the impact of statins on carcinogenesis places emphasis on genetic polymorphisms within these genes since cholesterol production genes represent the therapeutic targets of statin treatment. Additionally, it will be determined how the rs12916 HMGCR polymorphism, which codes for a protein that statins target, affects the chance of developing cancer, Orho Melander and colleagues [140] investigated a range of 26 to 41 single-nucleotide polymorphisms (SNPs) found in genes responsible for encoding proteins that play a role in cholesterol and its fraction synthesis. They confirmed the association between blood triglyceride levels and the risk of cancer as well as the potential association between the risk of breast and prostate cancer and the HMGCR polymorphism [133]. In contrast, the study connection was only marginally significant (OR 0.97; 95% CI 0.94–1.00; *p* = 0003) and did not link with the progression of proliferative alterations in males with prostate cancer (n = 22,773) compared to the control group (n = 23,050) [141]. Another study compared the risk of developing breast cancer (n = 122.977) in three HMGCR polymorphisms to the control group (n = 105.974). Breast cancer risk has not been evaluated to be significantly inhibited by HMGCR gene inhibition (OR 0.86; 95% CI 0.73–1.02; *p* = 0.09). Comparatively, statin users had a significantly lower risk of developing ovarian cancer (OR 0.60; 95% CI 0.43–0.83; *p* = 0.002 vs. OR 0.69; 95% CI 0.51–0.93; *p* = 0.01), whether they carried the BRCA1/2 mutation or not [142]. Functional genomics investigations must thus be taken into account since they reveal novel, possible therapeutic targets and aid in our understanding of the various clinical reactions experienced by patients taking the same medication, which helps to personalize care [136].

## 6. Conclusions

Statins work by blocking the enzyme HMG-CoA reductase, resulting in lower cholesterol levels and the prevention of cardiovascular disease as pharmacological effects. Beyond the lowering of lipoproteins, in recent years, there has been a burgeoning interest in delving into the molecular mechanisms that underlie the “pleiotropic” effects of statins. The most focus has been placed on statins’ antitumor properties, which have been supported by numerous preclinical studies. Here, we have included a summary of the most recent research on the antitumor mechanisms of statins. The first and most extensively researched statin-induced antitumor mechanism is the mevalonate pathway. Statins exert their influence on cell growth, proliferation, differentiation, and apoptosis by inhibiting the post-translational modification and activation of small GTPases and their subsequent signaling pathways. Additionally, the mevalonate pathway, crucial for cholesterol production, has been shown in various studies to have a significant correlation with cancer due to its role in regulating cellular cholesterol levels. As a result, we explored the role of the mevalonate pathway from both the cholesterol-mediated and non-cholesterol-mediated viewpoints in the antitumor actions of statins. Additionally, we emphasized the most current findings that provide additional insight into antitumor processes, such as autophagy, ferroptosis, tumor microenvironment targeting, and pyroptosis of statins. Also, many clinical and epidemiological studies on statins were discussed. Nonetheless, further in vitro studies are necessary to elucidate the mechanisms of statins, and clinical trials are needed to enhance our understanding of prevention strategies or potential cancer treatments based on statins.

## Figures and Tables

**Figure 1 cancers-15-04787-f001:**
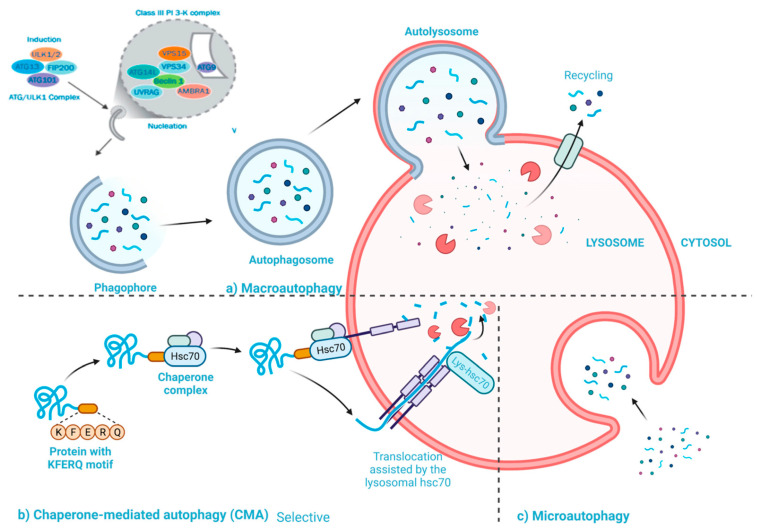
Schematic representation illustrating the three subtypes of autophagy [41].

**Figure 2 cancers-15-04787-f002:**
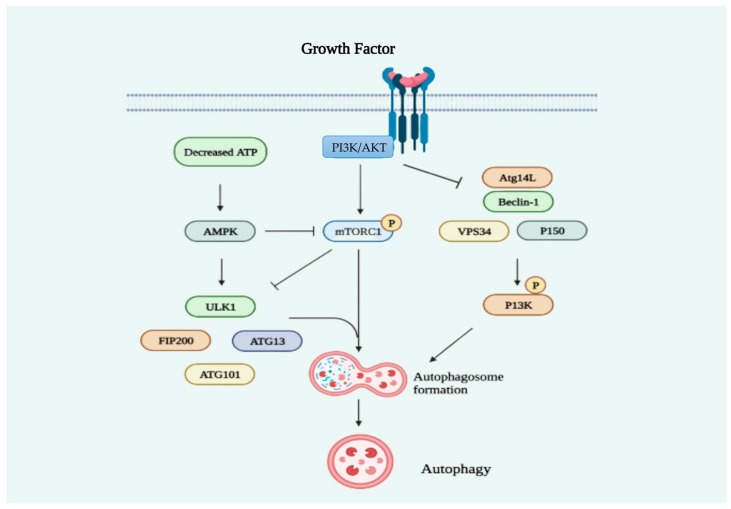
The principal signaling pathways implicated in the regulation of autophagy.

**Figure 3 cancers-15-04787-f003:**
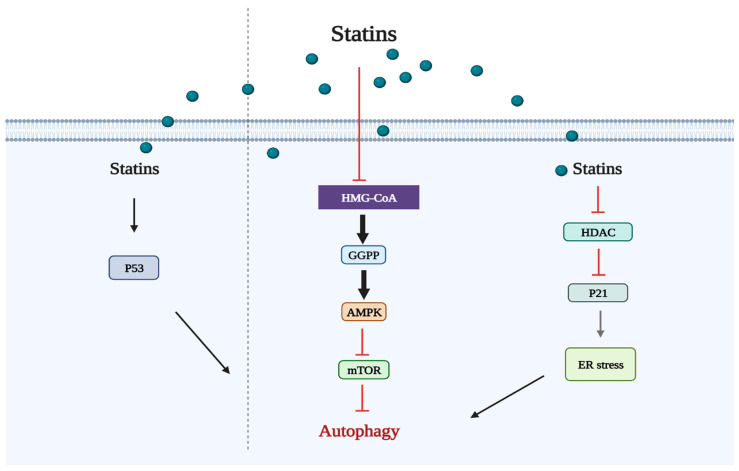
Autophagy signaling pathways triggered by statin use.

**Figure 4 cancers-15-04787-f004:**
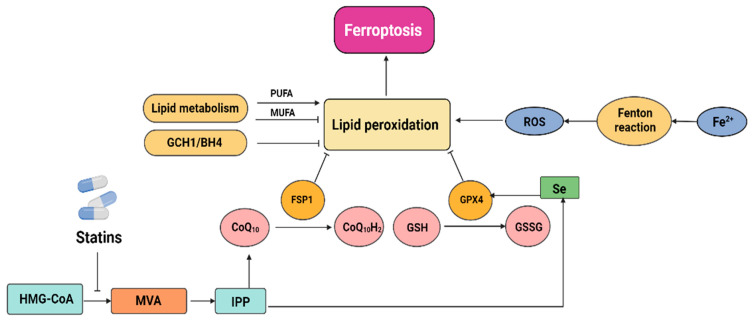
The interaction between ferroptosis and statins.

**Figure 5 cancers-15-04787-f005:**
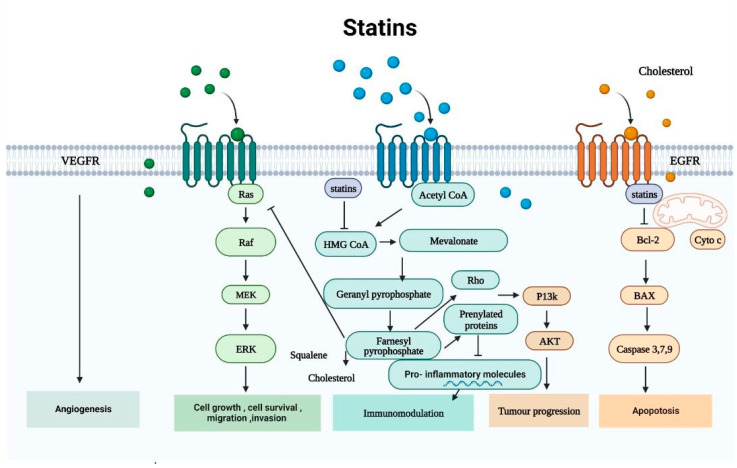
The use of statins as standalone anticancer treatments.

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
