# Peer review of "Unraveling the Anticancer Potential of Statins: Mechanisms and Clinical Significance"

_cancers, 2023, doi:10.3390/cancers15194787_

Round 1

Reviewer 1 Report

The manuscript is devoted to statins and summarizes their anticancer effects. The work is prepared very carefully, in a competent, very clear and understandable way. It introduces the reader to aspects of the anti-cancer mechanisms of statins. Notes the mevalonate pathway and non-cholesterol and cholesterol-mediated pathways. The role of statins in the regulation of autophagy, ferroptosis and pyroptosis is also discussed. The authors present the influence of statins on the microenvironment of the neoplastic tissue. They summarize the studies on the anticancer effect of statins in in vitro and in vivo studies. A large part of the slingshot is devoted to clinical trials of statins, in which they summarize the epidemiology, the impact of taking them on the incidence of cancer. They refer to publications based on meta-analysis, observational and pharmacological  studies, randomized controlled trials, mendelian randomization. The presented data were supported by well-selected literature (142 items).

After reading such an extensive study, I wanted to draw attention to the few editing errors as well as to discuss a few issues that, in my opinion, could improve the ease of reading and understanding the text.

Page 4,                                                                                                                                                                                     

The sentence - MiR-183 directly degrades ABCA1 mRNA to maintain high amounts of intracellular cholesterol, which helps colon cancer cells proliferate and have anti-apoptotic effects [33]. In the original publication, the authors use the spelling MiR-183, although the term miR-183 is currently used.

Page 6, 7 & 8                                                                                                                                                                    Letters A, B & C are used in the figure captions to represent autophagy subtypes (Figure 1), signaling pathways leading to autophagy (Figure 2) and their possible statin activation (Figure 3). Maybe it would be easier for the reader to read the figures if the letters A, B and C were also be included in the presented Figures?

Page 9                                                                                                                                                                                      

Line 3 and 4 from the bottom the sentence: Pyroptosis is a kind of inflammatory PCD that is distinct from ferroptosis and autophagy is pyroptosis. One of the pyroptosis repeated twice should be removed.

Page 10                                                                                                                                                                                  The sentence …….the long non-coding RNA (lncRNA) NEXNAS1/NEXN pathway. [76]. The dot before the reference number should be removed.

Page 11, 13                                                                                                                                                                                     

In my opinion, in vitro and in vivo should be written in italic. The more so that the authors do not use a unified notation.

Page 12 Paragraph 2 in the first line is HMC-CoA. Should it be HMG-CoA?

Page 14                                                                                                                                                                                      Line 5 in between …..40 mg/kg/day.Karim ….. should be a space. In the last sentence of the page, the cited work with number 118 was duplicated twice.

Page 18                                                                                                                                                                                       In references item 15, the title is given in capital letters.

Page 26                                                                                                                                                                           

At the end of the reference 1. should be deleted.

To make it easier for the reader to reach the abbreviations used in the text, I would suggest using a paragraph with abbreviations collected together in alphabetical order, if the requirements of the journal allow it.

In conclusion, I believe that the manuscript submitted for review is a very solid summary of the data available in the literature on the anti-cancer effect of statins and should be published after eliminating editing errors and the Authors' response to other discussion notes.

Author Response

Reviewer 1:

After reading such an extensive study, I wanted to draw attention to the few editing errors as well as to discuss a few issues that, in my opinion, could improve the ease of reading and understanding the text.

Page 4,                                                                                                                                                                                      

The sentence - MiR-183 directly degrades ABCA1 mRNA to maintain high amounts of intracellular cholesterol, which helps colon cancer cells proliferate and have anti-apoptotic effects [33]. In the original publication, the authors use the spelling MiR-183, although the term miR-183 is currently used.

Thanks for your comment. It has been corrected and highlighted in red in the revised manuscript.

Page 6, 7 & 8                                                                                                                                                                     

Letters A, B & C are used in the figure captions to represent autophagy subtypes (Figure 1), signaling pathways leading to autophagy (Figure 2) and their possible statin activation (Figure 3). Maybe it would be easier for the reader to read the figures if the letters A, B and C were also be included in the presented Figures?

Thank you for your comment. It has been corrected and highlighted in red in the revised manuscript.

Page 9                                                                                                                                                                                      

Line 3 and 4 from the bottom the sentence: Pyroptosis is a kind of inflammatory PCD that is distinct from ferroptosis and autophagy is pyroptosis. One of the pyroptosis repeated twice should be removed.

Thanks for your comment. It has been corrected and highlighted in red in the revised version.

Page 10                                                                                                                                                                                   The sentence …….the long non-coding RNA (lncRNA) NEXNAS1/NEXN pathway. [76]. The dot before the reference number should be removed.

Thanks for your comment. It has been corrected and highlighted in red in the revised version.

Page 11, 13                                                                                                                                                                                 

In my opinion, in vitro and in vivo should be written in italic. The more so that the authors do not use a unified notation.

Thanks for your comment. Based on the academic editor suggestion, he asked to use not italic and it has been corrected and highlighted in red color in the revised version.

Page 12 Paragraph 2 in the first line is HMC-CoA. Should it be HMG-CoA?

Thanks for your comment. It has been corrected and highlighted in red in the revised version.

Page 14                                                                                                                                                                                     

Line 5 in between …..40 mg/kg/day.Karim ….. should be a space. In the last sentence of the page, the cited work with number 118 was duplicated twice.

Thanks for your comment. It has been corrected and highlighted in red in the revised version.

Page 18                                                                                                                                                                                      In references item 15, the title is given in capital letters.

Thanks for your comment. It has been corrected and highlighted in red in the revised version.

Page 26                                                                                                                                                                                             At the end of the reference 1. should be deleted.

Thanks for your comment. It has been corrected and highlighted in red in the revised version.

To make it easier for the reader to reach the abbreviations used in the text, I would suggest using a paragraph with abbreviations collected together in alphabetical order, if the requirements of the journal allow it.

Thank you for your comment. In order to reduce the similarity percentage, we incorporated abbreviations in the text.

Reviewer 2 Report

In this review Authors have summarized the findings about the mechanisms of statins anticancer effects and their clinical implications.

The work appears complete and exhaustive, covering many different aspects of the implication of statins in cancer prevention, treatment, and sometimes, cell- and dose-dependent induction.

The most important mechanisms of this action have been mentioned and discussed, such as the induction of oxidative stress, cell cycle arrest, autophagy, and apoptosis of cancer cells.

My only criticisms concern the improvement of some figures.

I would increase the font size of some writings in figure 1 and/or change the color from light blue to black to make them more readable on a white background.

Fig.5; what are the green ligands on the left of the figure and the central receptors? Please specify better also in the legend.

Author Response

Reviewer 2:

In this review Authors have summarized the findings about the mechanisms of statins anticancer effects and their clinical implications.

The work appears complete and exhaustive, covering many different aspects of the implication of statins in cancer prevention, treatment, and sometimes, cell- and dose-dependent induction.

The most important mechanisms of this action have been mentioned and discussed, such as the induction of oxidative stress, cell cycle arrest, autophagy, and apoptosis of cancer cells.

My only criticisms concern the improvement of some figures.

I would increase the font size of some writings in figure 1 and/or change the color from light blue to black to make them more readable on a white background.

Thank you for your comment. Figure one is copyrighted, and I am not allowed to change it.

Fig.5; what are the green ligands on the left of the figure and the central receptors? Please specify better also in the legend.

Thank you for your comment. It represents the ligands of the cell receptors. We have specified this in the revised version.